# LucidAction: A Hierarchical and Multi-model Dataset for Comprehensive Action Quality Assessment

Linfeng Dong[1,2], Wei Wang[2], Yu Qiao[2], and Xiao Sun[2]

[1]Zhejiang University
[2]Shanghai Artificial Intelligence Laboratory
{donglinfeng, wangwei, sunxiao}@pjlab.org.cn, yu.qiao@siat.ac.cn

## Abstract

Action Quality Assessment (AQA) research confronts formidable obstacles due to limited, mono-modal datasets sourced from one-shot competitions, which hinder the generalizability and comprehensiveness of AQA models. To address these limitations, we present LucidAction, the first systematically collected multi-view AQA dataset structured on curriculum learning principles. LucidAction features a three-tier hierarchical structure, encompassing eight diverse sports events with four curriculum levels, facilitating sequential skill mastery and supporting a wide range of athletic abilities. The dataset encompasses multi-modal data, including multi-view RGB video, 2D and 3D pose sequences, enhancing the richness of information available for analysis. Leveraging a high-precision multi-view Motion Capture (MoCap) system ensures precise capture of complex movements. Meticulously annotated data, incorporating detailed penalties from professional gymnasts, ensures the establishment of robust and comprehensive ground truth annotations. Experimental evaluations employing diverse contrastive regression baselines on LucidAction elucidate the dataset's complexities. Through ablation studies, we investigate the advantages conferred by multi-modal data and fine-grained annotations, offering insights into improving AQA performance. The data and code will be openly released to support advancements in the AI sports field.

## 1 Introduction

The comprehensive evaluation of human actions, capturing both their strengths and weaknesses as well as the quality of their execution, finds extensive applicability in various fields. This is exemplified by AI-powered fitness applications that deliver customized workout regimes [7, 39, 12, 22, 38]. Notably, the 2020 Tokyo Olympics pioneered the use of AI in gymnastics scoring, enhancing both fairness and precision in evaluations [1]. Additionally, motion gaming systems employ sophisticated assessments of user actions to create immersive and interactive experiences [18, 21, 27]. The influence of this task spans diverse industries, including education, sports, and entertainment. As technological advancements continue, the impact of such evaluations is expected to grow significantly.

Prior research [35, 32, 31, 33, 37] has raised the task of Action Quality Assessment (AQA) in tackling the issue of human action evaluation, aiming to regress a definitive quality score for the performed action directly. Unlike action recognition [17], which assumes consistency within the same action type, AQA is inherently more challenging as it must discern subtle variations in action execution

Submitted to the 38th Conference on Neural Information Processing Systems (NeurIPS 2024) Track on Datasets and Benchmarks. Do not distribute.

quality, including swiftness, intensity, and timing, among performers. Additionally, AQA lacks clearly defined quality metrics and requires expertise for evaluation. Given these formidable challenges, the quantity, professionalism, and diversity of high-quality AQA datasets significantly lag behind those of action recognition datasets, severely impeding the advancement of AQA research.

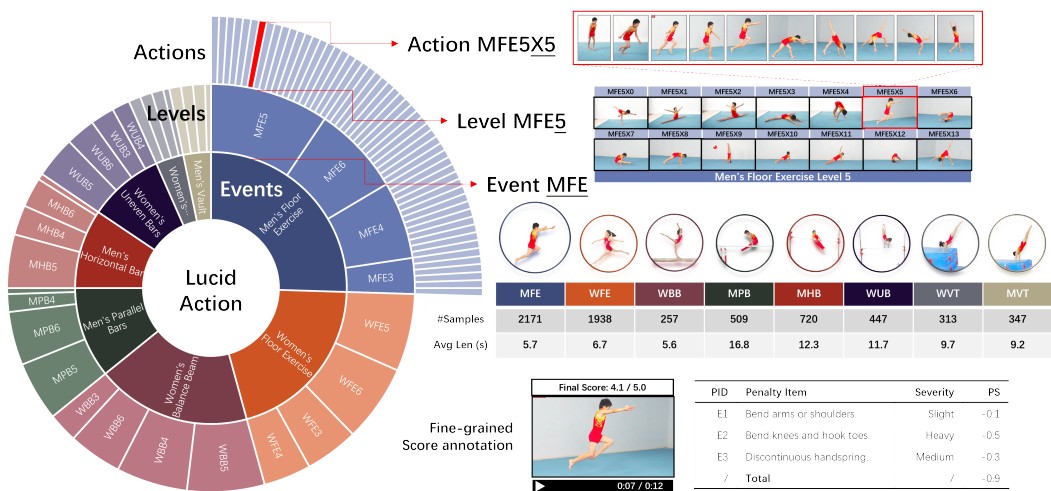

Figure 1: An overview of the LucidAction dataset. LucidAction adopts a three-tier hierarchical structure of Sport Events, a first-introduced concept "Curriculum Levels" and Actions. It provides a diverse range of actions and detailed penalty-based score annotation to seek better comprehensibility in action quality assessment.

To facilitate this research, a few datasets [35, 31, 33, 45, 47] – gathered primarily from web sources – have been introduced. These datasets predominantly consist of video footage of individual sports competitions like diving or skating, sourced from various sports television broadcasting, such as the Olympic Games, and paired with the corresponding judges' scores. Unfortunately, due to the nature of the data sources, the AQA models trained on these datasets are limited to application in a 'one-shot examination' that represents the highest level of a sport. As a result, they cannot be widely utilized by general enthusiasts and learners, significantly narrowing their scope and frequency of use. Moreover, mono-modal input of video captured by a single moving camera [31, 33, 47] and the absence of a detailed scoring process for the final score severely curtail the model's adaptability and comprehensibility in diverse data settings.

*Humans and animals learn much better when the examples are not randomly presented but organized in a meaningful order which illustrates gradually more concepts, and gradually more complex ones.*
*– Curriculum Learning, Yoshua Bengio et.al.*

To surmount the limitations of current action assessment research, we introduce LucidAction, the first AQA dataset structured according to the principles of curriculum learning. LucidAction introduces a curriculum-based approach to organize data, aligning with the natural learning progressions observed in sports training. It comprises a three-tier hierarchical structure, including eight diverse sports events and four difficulty levels for each event. This hierarchical structure facilitates sequential skill acquisition and accommodates a wide spectrum of athletic abilities. Additionally, the dataset harnesses a high-precision multi-view Motion Capture (MoCap) system to capture complex movements accurately. It integrates 2D pose estimation and multi-view triangulation to acquire precise 3D pose annotations. Furthermore, the dataset includes annotations by professional gymnasts, ensuring the provision of robust and comprehensive ground truth data for AQA models. Through rigorous experimentation, we investigate the effectiveness of multi-modal inputs and fine-grained hierarchical annotations in enhancing AQA performance, thereby offering insights into methodological advancements for the field.

## 2 Related Work

In this section, we provide a concise overview of previous AQA datasets and methodologies.

Table 1: Comparison of LucidAction and existing action quality assessment datasets. #Sport is number of the sport event in dataset, e.g. diving, figure skating, etc. In Anno.Type, S indicates coarse-grained action score, PS indicates progress-aware penalty-based score annotation. In Modality, V, T, A, P indicate video, text, audio, pose.

| Dataset | Year | #Sport | Source | Anno.Type | Modality | #Sample | #Level | #Action | #View |
|---|---|---|---|---|---|---|---|---|---|
| MIT Dive&Skate [35] | 2014 | 2 | web | S | V | 309 | 1 | - | 1 |
| UNLV Dive&Valut [32] | 2017 | 2 | web | S | V | 546 | 1 | - | 1 |
| AQA-7 [31] | 2019 | 7 | web | S | V | 1189 | 1 | - | 1 |
| MTL-AQA [33] | 2019 | 1 | web | S | V, T | 1412 | 1 | 58 | 1 |
| FisV [45] | 2019 | 1 | web | S | V | 500 | 1 | - | 1 |
| FSD-10 [24] | 2020 | 1 | web | S | V | 1484 | 1 | - | 1 |
| Rhythmic Gymnastics [51] | 2020 | 4 | web | S | V | 1000 | 1 | - | 1 |
| FR-FS [41] | 2021 | 1 | web | S | V | 417 | 1 | - | 1 |
| FS1000 [42] | 2022 | 1 | web | S | V, A | 1604 | 1 | - | 1 |
| FineDiving [47] | 2022 | 1 | web | S | V | 3000 | 1 | 52 | 1 |
| OlympicFS [11] | 2023 | 1 | web | S | V, T | 200 | 1 | - | 1 |
| RFSJ [25] | 2023 | 1 | web | S | V | 1304 | 1 | - | 1 |
| **LucidAction (Ours)** | 2024 | 8 | mocap | S, PS | V, P | 6702 | 4 | 259 | 8 |

**Action Quality Assessment Datasets.** Existing AQA datasets cover various domains like diving [35, 32, 31, 33, 47], figure skating [32, 45, 41, 25, 24, 42, 11], gymnastic [32, 51] and other general sports [4, 34, 53]. As shown in Table 1, previous datasets typically provide RGB videos with video-level scores from multiple judges. Despite the human-centric nature of AQA, none incorporate pose data. Only a few AQA approaches [35, 30, 29] consider extracting 2D pose feature from mono-view video. It is likely due to the difficulty of reliable pose estimation from fast motions in mono-view video captured by moving camera. Another key attribute of AQA datasets is the annotation of action score given by experts under guideline of sport-specific scoring rules. Earlier datasets such as AQA-7 [31] contained only overall scores and sport classes, while MTL-AQA [33] provide fine-grained action type and transcribed video commentary as language modality. FineDiving [47] introduced a two-level annotation with action classes and fine-grained subclasses to capture action procedures, but without procedure-aware scores. FS1000 [42] expanded annotations along five quality aspects. A key challenge has been the laborious collection and annotation of such fine-grained data, requiring collaboration of players, coaches, and referees. Thus, existing datasets focus on top athletes in competitions from web sources, neglecting the skill development processes from practice. In summary, current AQA datasets are limited by: (1) lacking pose modality, (2) coarse annotations without step-wise scores, (3) a focus on elite rather than progressive skill acquisition. Our proposed LucidAction dataset is the first to provide both RGB and 3D pose, with richer annotations and technical skills than previous datasets.

**Action Quality Assessment.** Currently, AQA approaches mainly follow three formulations: **1)** *Direct regression* formulation supervised by score is widely used in sports AQA approach [35, 32, 43, 30, 31, 51, 29, 33, 34, 45, 37, 41, 44]. Some approaches perform segmentation [52, 26] or localization [15, 13] to generate subaction sequence and predict subscore for each subaction. Recent works incorporate auxiliary input, including music [42], language commentary [11], group formation[53] to improve their ability in AQA. **2)** *Pairwise ranking* is adopted in daily-life AQA [9, 10, 20] or specific sport scenario [4] where precise executing score of action is not available. These approaches mainly focus on overall ranking, limiting their application when requiring quantitative action analysis. **3)** *Pairwise regression* formulation [19, 25] is first proposed by Siamese Network [14] and CoRe [50] to learn the relative score by pair-wise comparison. TPT [3] adopt learnable queries as positional encoding to decode action sequences into a fixed number of temporal-aware part representations. TSA [47] explicitly segment action sequence into consecutive steps and apply procedure-aware cross-attention between target and exemplar corresponding steps.

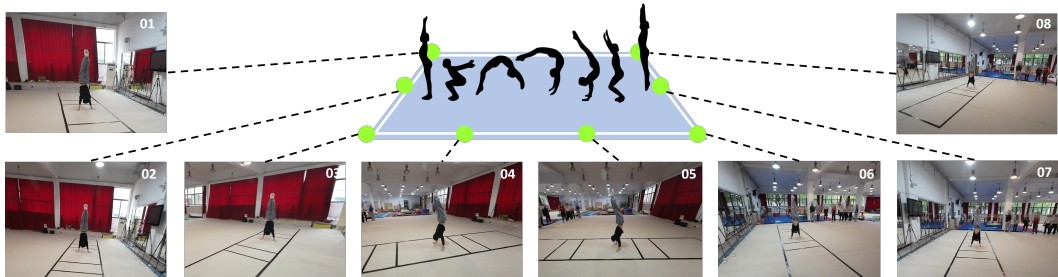

Figure 2: Camera layout and corresponding frames for event MFE, please refer to the supplementary materials for camera layouts of other events.

## 3  The LucidAction Dataset

The acquisition and refinement of specific sporting skills by individuals constitute a multifaceted process. Typically, it entails initial engagement in specialized exercises aimed at fostering fundamental abilities, which are systematically deconstructed into simpler components. Building upon this foundational framework, further progress is achieved through the adept and strategic amalgamation of these movements to accomplish more intricate objectives in sports competitions.

In order to closely mirror this natural progression of skill acquisition observed in curriculum learning, we have structured our dataset based on the official teaching curriculum outlined in the *Regulations on the Movement and Scoring Standards of Chinese Gymnastics Sports Levels* (*Standards* for brevity), as promulgated by the Chinese Gymnastics Association. The adoption of the *Standards* is particularly advantageous due to its widespread utilization in local sports instruction and grading examinations, facilitating the organization of proficient athletes and instructors and the subsequent collection of corresponding sports and assessment data.

As depicted in Figure 1, we introduce a three-tier hierarchical structure. Notably, for the first time, we incorporate the concept of sports "Curriculum Levels" into our dataset. (1) ***Sports Event.*** We offer the most diverse range of sports events to date - 8 in total, namely men's/women's floor exercise (MFE, WFE), vault (MVT, WVT), men's parallel bars (MPB), horizontal bars (MHB), women's uneven bar (WUB), balance beam (WBB). (2) ***Curriculum Level.*** Each sports event within our dataset encompasses four distinct levels of difficulty, ranging from easy to challenging. This pioneering inclusion of difficulty levels within an AQA dataset establishes the cornerstone of our proposed LucidAction benchmark. In educational contexts, learners typically progress through these levels sequentially, demonstrating mastery and passing assessments at each stage before advancing. This methodology not only furnishes a rich, multi-tiered dataset conducive to AQA model training but also accommodates a diverse spectrum of athletic abilities. (3) ***Actions.*** Within each curriculum level, a collection of representative actions is delineated, with each action type constituting a movement routine lasting an average of 8.6 seconds, serving as the finest-grained unit of analysis. On average, each curriculum level comprises 65 representative actions, culminating in a total of 259 actions across all levels and events.

### 3.1  Multi-View Motion Capture and Multimodality

We deploy a high-precision Motion Capture (MoCap) system. The cameras used in this system are DJI Osmo Action 3 and work in the mode of 4096×4096 (4K) resolution and 60fps. Temporal and spatial calibrations between multiple cameras are performed using standard tools [28, 2].

**Multi-View and High Spatiotemporal Resolution.** For gymnastics events, a variety of poses including lying, crouching, rolling up, and rapid jumping are performed, involving significant self-occlusion and swift movements. These complex scenarios bring considerable challenges in accurately inferring 3D poses from conventional single-view RGB or depth sensors, greatly impacting AQA performance. To tackle this issue, we established the first multi-view (8 views in total) MoCap system

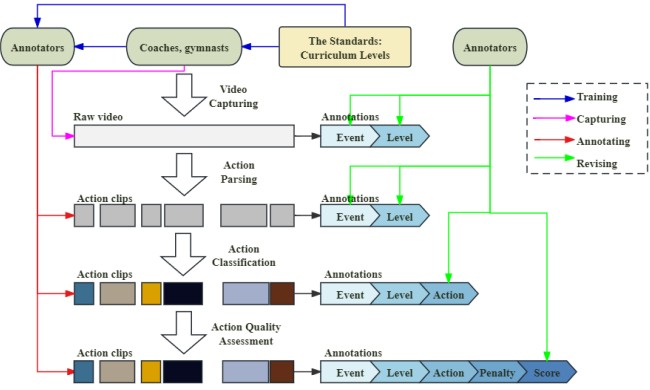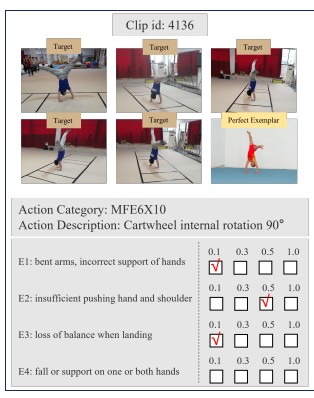

(a) Annotation Pipeline. The video capturing process is scheduled by *The standards*. Left shows the action clips, right shows the corresponding hierarchical labels. All annotators are trained by professional coaches and gymnastics with code of points in *The standards* before annotation.

(b) Annotation tool assessment system layout, annotators can compare the target action clip with perfect exemplar from all eight camera views.

Figure 3: Illustration of annotation pipeline and system layout.

with high-quality (4K, 60fps) video output tailored for the AQA task. Our experiments confirm the significant performance enhancement brought by leveraging multi-view video information for the AQA task. Figure 2 illustrates the camera layout and corresponding multi-view frames of Men's/Women's Floor Exercise in our LucidAction Dataset. Illustrations of other sport events can be found in supplementary materials. The release of the dataset obtained consent from all athletes appearing in the videos. We employ facial anonymization algorithm deface [48] to protect the sensitive identity information of the athletes.

**Multi-Modality for Diverse Applications.** We attain high-precision 3D pose annotations by multi-view 2D pose estimation and 3D pose reconstruction. We used a hybrid 2D pose estimation approach involving both algorithms and human review in three stages: (1) We employed RTMpose [16] pretrained on 7 public datasets to estimate 2D poses from single-view videos followed by human quality checks. In this stage, estimated 2D on some action categories may fail human review due to their rare appearance in the pretraining datasets; (2) We manually annotated 2D poses of these failed actions, fine-tuned the RTMpose model, and re-estimated the 2D poses, which were then reviewed again; (3) Any 2D poses that still failed the review were manually annotated. This approach balances automated efficiency with human validation to ensure accurate 2D pose groundtruth. For 3D pose estimation, we reconstructed 3D poses using multi-view 2D poses as groundtruth, a common method in creating 3D pose datasets [36, 23, 5, 8]. Reconstructed 3D pose from multi-view 2D are accepted as groundtruth in tasks like human action recognition [40] and motion prediction [46]. Follow these works, we assess that the accuracy of our 3D poses reconstruction pipeline is sufficient for the AQA task. To gauge the accuracy of the automatic pose annotation pipeline, we manually annotate a subset of data. In the experiments, we thoroughly compare the performance of AQA models across different modalities.

## 3.2 Data Annotation

We provide professional, comprehensive and reliable ground truth annotations in the LucidAction dataset for the action quality assessment task.

**Hierarchical Actions Construction** We employ a multi-stage strategy to gather extensive hierarchical action labels based on inherent levels (Sports Event, Curriculum Level, and Action). The annotation process is depicted in Figure 3a. Raw videos are systematically captured according to predefined standards, with planned recording sessions for sports events and curriculum levels. As a result, each raw video inherently includes annotations for the first two hierarchies at the time of recording. When

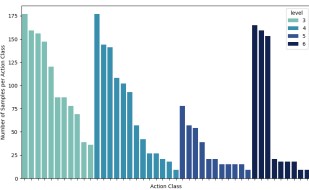
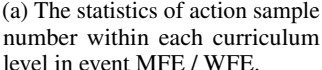
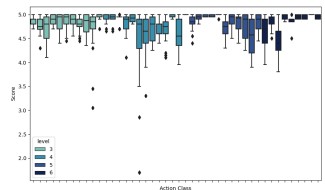
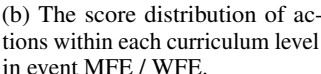
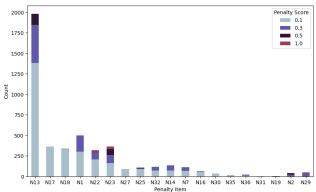

(a) The statistics of action sample number within each curriculum level in event MFE / WFE.

(b) The score distribution of actions within each curriculum level in event MFE / WFE.

(c) The statistics of penalty items and penalty score appear in event MFE / WFE.

Figure 4: The statistics of action samples, scores and penalties.

dealing with raw videos containing multiple actions, ten annotators first segment them into slices containing only one action. Subsequently, they assign the action category of each slice based on the corresponding sports event and curriculum level.

**Professionalism and Robustness** We enlist the expertise of professional gymnasts, referees, and coaches to aid us in action sequences collection and score annotation. We conducted a five-month data capturing during professional gymnastics training courses organized according to *the Standards* at a sports university. To ensure the annotation quality and reduce potential subjective bias, all annotators have taken classes from referees on how to score action according to *the Standards*. To further mitigate bias, each action segment is assessed by at least five annotators repeatedly. To avoid neglecting errors due to view occlusion, action footage from all views are provided to the annotators.

**Detailed Penalty Items Annotation.** Previous efforts solely yielded a final scoring outcome without disclosing the intricacies of the scoring process, thus deviating from the authentic assessment procedure and compromising result comprehensibility. In a pioneering move, we provide comprehensive annotations detailing the scoring process. For each action, the execution quality is evaluated, according to *the Standards*, by identifying up to 5 specific penalty items, each indicates a possible execution error. For each penalty item, we assess whether the corresponding error occurs in the action, and based on the severity of the error from light to heavy, assign a penalty score from {0.1, 0.3, 0.5, 1.0}. The statistics of score and penalty items are shown in Figure 4.

## 4 Experiment

In this section, we will demonstrate how LucidAction will substantiate the objectives of comprehensive AQA through three key dimensions: contrastive regression workflow, multi-model input and fine-grained hierarchical annotations.

### 4.1 Contrastive Regression Workflow

Fundamentally, the assessment of an action must considers the context of a particular sports scenario, as it requires attention to sports-specific goals and metrics. For example, although both activities entail running, the technical standards for a 100-meter sprint and a football match can diverge significantly. Therefore, AQA inherently demands an in-context mechanism employing exemplars for the contextual calibration of assessments, eschewing an absolute valuation of the action.

We embrace the recently established pair-wise contrastive regression approaches Siamese Network [14], CoRe [50], TSA [47] and TPT [3] as main baseline architecture, concisely encapsulated within the framework illustrated in Figure 5. This architecture consists of four interconnected modules, (1) a **backbone** $\mathcal{B}$ to encode input signals into deep network features; (2) an **action decoder** $\mathcal{A}$ to extract key motion features across temporal dimension; (3) a **pair encoder** $\mathcal{P}$ to facilitate interactions between targets and exemplars for contrastive purposes; (4) a **score regressor** $\mathcal{S}$ to map interaction features into relative scores. Given a pairwise target $X$ and exemplar $Z$, the the contrastive regression

problem can be represented as:

$$\hat{y}_X = \mathcal{S}(\mathcal{P}(\mathcal{A}(\mathcal{B}(X)) \oplus \mathcal{A}(\mathcal{B}(Z))) \mid \Theta) + y_Z \qquad (1)$$

where $\Theta$ indicates the learnable parameters, $\hat{y}_X$ is the predicted score of target $X$, $y_Z$ is the ground-truth score of exemplar $Z$, $\oplus$ denotes the operation to fuse the target and exemplar's representations after the action decoder. In experiments we use concatenation following previous work TPT [3].

We compare the results of contrastive regression baselines and a direct regression approach USDL[37] on our newly proposed benchmark LucidAction. We also list the baseline performance on three publicly available datasets AQA-7 [31], MTL-AQA [33], FineDiving [47] as reference (see the supplement for more details on these datasets).

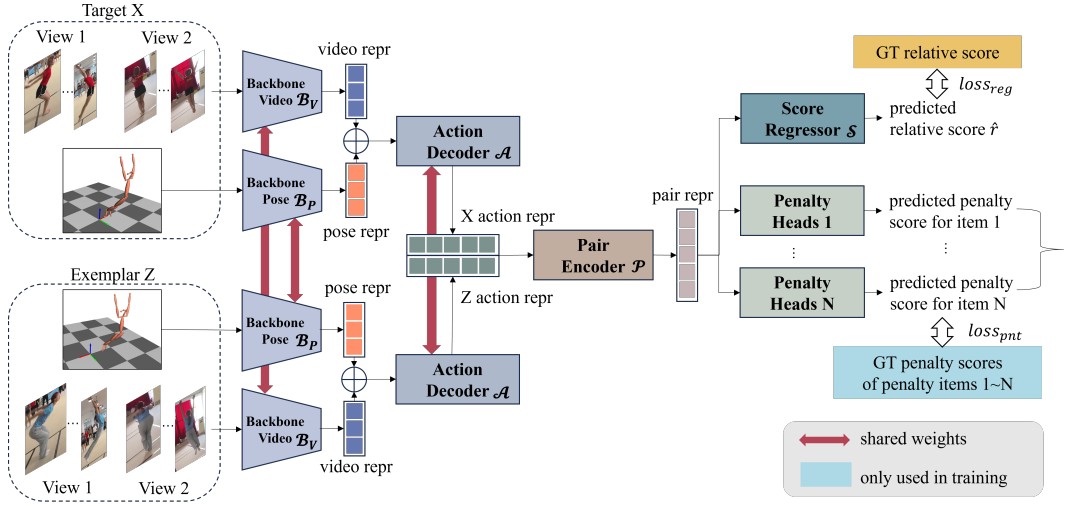

Figure 5: An overview of contrastive regressive workflow with additional penalty heads.

**Implementation Details.** We adopt I3D pretrained on Kinetics [6] as video backbone for all baselines. TPT [3] uses a 2-layer transformer block as action decoder, a 2-layer MLP as pair encoder and another 2-layer MLP as score regressor. We extract 103 frames for each video or pose sequence and stack them with interval 5 as 20 clips, each contains 8 frames. For More implementation details on other baselines, data augmentation, learning rate, training epoch, optimization, inference, and so on, please refer to the supplementary materials.

**Evaluation Metrics.** To facilitate comparison with previous work in AQA [35, 31, 37, 41, 47], we employ two metrics in our experiments: Spearman's rank correlation ($\rho$) and relative L2 distance(R-$\ell_2$). Spearman's rank correlation assesses the rank correlation between predictions and ground-truth scores, The relative L2 distance focuses on the numerical scoring difference between predictions and ground-truth scores.

Table 2: Baseline performance comparison on LucidAction and former AQA datasets.

| Method | AQA-7 | | MTL-AQA | | FineDiving | | LucidAction | |
|---|---|---|---|---|---|---|---|---|
| | $\rho \uparrow$ | R-$\ell_2(\times 100) \downarrow$ | $\rho \uparrow$ | R-$\ell_2(\times 100) \downarrow$ | $\rho \uparrow$ | R-$\ell_2(\times 100) \downarrow$ | $\rho \uparrow$ | R-$\ell_2(\times 100) \downarrow$ |
| USDL[37] | 0.810 | 2.57 | 0.923 | 0.468 | 0.891 | 0.382 | 0.540 | 0.708 |
| CoRe [50] | 0.840 | 2.12 | 0.951 | 0.260 | 0.906 | 0.362 | 0.625 | 0.685 |
| TSA [47] | 0.848 | 2.07 | 0.947 | 0.284 | 0.920 | 0.342 | 0.643 | 0.690 |
| TPT [3] | **0.872** | **1.68** | **0.960** | **0.238** | **0.945** | **0.218** | **0.701** | **0.624** |

**Baseline Model Results.** The baseline performance on LucidAction and the established dataset, namely AQA-7, MTL-AQA and FineDiving, is summarized in Table 2. Contrastive regression methods significantly outperforms direct regression across all four datasets. On LucidAction, the best-performing TPT model improves $\rho$ that evaluates model's relative scoring ability by 30% and R-$\ell_2$ that

evaluates the absolute scoring ability by 12% compared to USDL. Contrastive regression approaches empower models to focus on visual disparities that frequently encapsulate crucial scoring information between target and exemplar, thereby effectively filtering out extraneous noise such as background interference and attire variation. Furthermore, the contrastive regression approach enhances data utilization by furnishing multiple exemplars for a single target action, thereby generating diverse paired inputs. This diversification enriches the evaluation process, augmenting the robustness of the assessment results. Given the superior performance achieved by TPT across all four datasets as delineated in Table Table 2, we adopt TPT variants for subsequent ablation studies.

## 4.2 Multi-model Input

We employ unified network architectures, loss functions, and training methods across different data modalities to ensure a fair comparison. The only difference lies in using ST-GCN [49] pre-trained on NTU RGB+D[36] as backbone for pose sequence input, as illustrated in Figure 5.

**Multi-view RGB Video Data.** To investigate the potential benefits of incorporating multi-view RGB videos, we conduct two multi-view stategies. Batch strategy puts different views in batch dimension as separate samples, while the channel strategy places different views on channel dimension within one sample. We also investigate the effects of channel fuse position (Pos) and operation (Opt), namely concatenation (*Cat*) and averaging (*Avg*). For experimental settings, multi-view test setting (Mv.Test) utilizes multi-view inputs during both training and testing phases, while the single-view test setting (Sv.Test) employs multi-view input only during training and duplicates single-view input during testing to simulate real-world scenarios where multi-view data may not be available. For further model details, please refer to the supplementary materials.

Table 3: Ablation studies of multi-model inputs.

(a) Multi-view ablation.

| Strategy | Pos | Opt | Mv.Test | Sv.Test |
|---|---|---|---|---|
| *Base* | - | - | - | 0.701 |
| *Batch* | - | - | - | 0.730 |
| | *BB* | *Cat* | 0.736 | 0.729 |
| | | *Avg* | 0.724 | 0.712 |
| | *AD* | *Cat* | 0.742 | 0.726 |
| *Channel* | | *Avg* | 0.737 | 0.728 |
| | *PE* | *Cat* | **0.759** | **0.747** |
| | | *Avg* | 0.713 | 0.703 |
| | *SR* | *Avg* | 0.732 | 0.730 |

(b) Pose modality ablation. When using dual-stream, the feature extracted by I3D and ST-GCN are concatenated before action decoder.

| Data Modality | $\rho \uparrow$ | R-$\ell_2(\times 100) \downarrow$ |
|---|---|---|
| *RGB* | 0.701 | 0.624 |
| *Pose2d* | 0.605 | 0.898 |
| *Pose3d* | 0.689 | 0.593 |
| *RGB+Pose3d* | **0.746** | **0.560** |

As depicted in Table 3a, introducing multi-view on batch to increase training data results in a $4.1\%$ improvement from 0.701 to 0.730. Multi-view input on channel yields a slightly higher performance than batch in Mv.Test and comparable performance in Sv.Test, except for concatenation after the *Pair Encoder* that gains a $6.6\%$ improvement from 0.701 to 0.747. This enhancement can be attributed to the capability of capturing errors obscured in a single view and leveraging implicit 3D knowledge, including depth information and shared objects across two synchronized views. Concatenation outperforms averaging in most positions since averaging causes information loss.

**Human Pose Data** We explore the impact of using different input modalities—2D human body pose, 3D human body pose, and RGB-pose dual-stream—on the AQA task. We observe in Table 3b that using only 2D poses reduces the model's performance on correlation $\rho$ from 0.701 to 0.605, using only 3D poses yields a correlation performance of 0.689, slightly lower than RGB input, but with an improved R-$\ell_2$ from 0.624 to 0.593. The decrease may stem from the abstract nature of keypoint data, leading to a loss of crucial information for action assessment. Conversely, combining dual-stream inputs with RGB and 3D poses results in a 6.4% improvement on $\rho$ from 0.701 to 0.746. One potential explanation is that human pose data is more conducive to the model in comparing key kinematic properties of the target and exemplar, such as keypoint movement velocity, displacement distance, angles, etc.

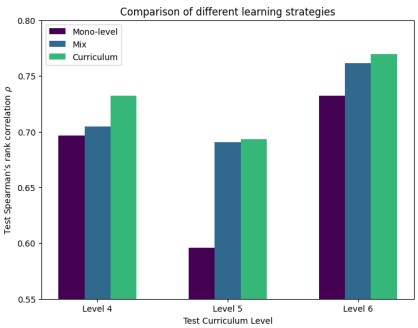

Figure 6: Comparison of different learning strategies.

| #Penalty Head | $\rho \uparrow$ | R-$\ell_2(\times 100) \downarrow$ |
|---|---|---|
| 0 | 0.701 | 0.624 |
| 1 | 0.733 | 0.539 |
| 2 | **0.741** | 0.514 |
| 3 | 0.735 | **0.501** |

Table 4: Ablation study of the number of penalty items used as additional supervision only during training.

### 4.3 Fine-grained Hierarchical Annotations

LucidAction is presented with a curriculum hierarchy and fine-grained penalty labels for scoring. In this section, we study whether these annotations help model's understanding of action quality.

**Curriculum Level.** We investigate the impact of curriculum level on the AQA task through two training methods: 1) *Mixed learning*, which trains on a shuffled LucidAction dataset with all levels; and 2) *Curriculum learning*, which organizes training data by level order, gradually introducing more difficult actions and complex quality concepts. Additionally, we compare models trained on individual levels. Analysis presented in Figure 6 demonstrates that models trained with mixed levels outperform those trained on a single level for any test level. This is particularly evident for level 5 actions, where fewer samples are available, indicating the model's ability to learn universal action quality concepts across different levels. Moreover, when utilizing the same volume of training data, curriculum learning surpasses mixed learning across all levels. This validates our hypothesis that the gradual progression of curriculum learning facilitates the development of complex quality concepts upon simpler ones learned earlier.

**Detailed Penalty Items.** The inclusion of unique penalty item annotations in LucidAction enhances the comprehensiveness and reliability of score annotations. In our experiments, we assess the benefits of incorporating this supervision. As illustrated in Figure 5, we introduce a plug-and-play multi-head network, each head corresponds to a binary classification auxiliary tasks, identifying whether the execution errors specified by a penalty item occur (penalty value > 0). Specifically, we focus on the three most frequent penalties N12, N17 and N18 in Figure 4c. Results in Table 4 indicate that models augmented with penalty heads achieve notable improvements, with correlation ($\rho$) increasing up to 0.741 (+5.7%) and R-$\ell_2$ up to 0.501 (+20%). This suggests that fine-grained penalty labels enhance the model's understanding of action quality. Additionally, the adoption of penalty-based annotation enables intentional collection of penalty-free samples for each action category, ensuring the availability of perfect exemplars. If no perfect action is captured during regular training sessions, specialized gymnasts will perform additional recordings to ensure each action category includes a perfect sample. Perfect exemplars are challenging to obtain in previous datasets [31, 33, 47] collected from one-shot public competitions. However, in our work, if no perfect action is captured during regular training sessions, specialized gymnasts will perform additional recordings to ensure each action category includes a perfect sample. Further ablation experiments regarding exemplar quality and quantity are presented in the supplementary materials.

## 5 Limitations and Other Applications

**Limitations.** LucidAction is gathered within controlled environments utilizing a high-precision multi-view Motion Capture (MoCap) system. However, it may not fully replicate real-world conditions where variables such as lighting, background, and other environmental factors can significantly vary.

Despite annotations being provided by professional gymnasts, subjective biases during scoring may still exist. Ensuring consistent and objective annotations remains a challenge.

**Applications.** LucidAction offers distinct advantages for motion generation, particularly due to the structured and standardized nature of gymnastics movements, which reduces ambiguities often encountered in daily actions. LucidAction can be utilized to develop educational tools and simulations that teach gymnastics techniques, providing proper form and execution, aiding in skill development.

## 6   Conclusion

In this paper, we introduce LucidAction, a novel dataset designed for Action Quality Assessment (AQA) featuring a hierarchical structure with eight diverse sports events and four curriculum levels. Leveraging a high-precision multi-view Motion Capture (MoCap) system, LucidAction offers rich and comprehensive data including multi-view RGB video, 2D and 3D pose for action assessment. Through experimentation with contrastive regression baselines on LucidAction, we have demonstrated the efficacy of multi-modal input and fine-grained annotations in enhancing AQA tasks. We anticipate that the LucidAction dataset, alongside our experimental findings, will serve as valuable resources for researchers and practitioners within the field of action quality assessment.

**Acknowledgements.** The work is supported by the National Key R&D Program of China (No. 2022ZD0160104).

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
