# OpenReview forum: "LucidAction: A Hierarchical and Multi-model Dataset for Comprehensive Action Quality Assessment"
_NeurIPS.cc/2024/Datasets_and_Benchmarks_Track — NeurIPS 2024 Track Datasets and Benchmarks Poster_

### Official Review · Reviewer_FFMn · 2024-06-29
**LucidAction: A Hierarchical and Multi-model Dataset for Comprehensive Action Quality Assessment**

**Rating:** 6
**Confidence:** 3
**Correctness:** The claims are correct, and is constr…
**Clarity:** yes

**Review:**

LucidAction stands out as a meticulously curated Action Quality Assessment (AQA) dataset, designed to address critical gaps in current data resources. This dataset is a valuable asset for researchers and professionals in the AQA domain, offering a rich repository for advancing the field.
The paper detailing LucidAction's creation is thorough and insightful, providing a clear exposition of its rationale, methodology, and outcomes. The dataset's hierarchical organization, multi-modal data inclusion, and comprehensive annotations are described with precision, while the experimental results are presented in a structured and accessible format.
LucidAction introduces innovative features that set it apart from conventional AQA datasets. Its comprehensive nature and meticulous structuring promise to propel AQA research and practical applications forward. By facilitating the development of more precise and dependable AQA models, LucidAction opens up new possibilities in diverse sectors, including sports coaching, fitness technology, and interactive gaming.
Pros:
1. The hierarchical structure and diverse sports events provide a rich and varied dataset for AQA research.
2. Multi-view RGB video and 3D pose annotations allow for accurate and detailed analysis of action quality.
3. Fine-grained penalty-based score annotations enhance the understanding of action quality and enable the development of more robust AQA models.
Cons:
1. The dataset was collected in controlled environments, which may limit its applicability to real-world scenarios with varying lighting, backgrounds, and other environmental factors.
2. Despite annotations by professional gymnasts, subjective biases during scoring may still exist.
3. The dataset currently focuses on gymnastics events, and expanding to other sports would further enhance its applicability.

**Strengths:**

Researchers have introduced a systematically collected AQA dataset structured on curriculum learning principles. The dataset features a three-tier hierarchical design, encompassing eight diverse sports events across four curriculum levels, facilitating a sequential skill acquisition process and accommodating a broad range of athletic abilities. Moreover, LucidAction incorporates multi-modal data, including multi-view RGB video, 2D and 3D pose sequences, along with meticulous penalty details from professional gymnasts, ensuring the accuracy and comprehensiveness of annotations.
The creation of LucidAction enriches the resources for AI-powered fitness applications, sports scoring, and motion gaming systems, driving advancements in education, sports, and entertainment industries. As technology evolves, the significance of such evaluative methodologies will continue to grow.
Regarding research quality, experimental outcomes evaluated LucidAction using a variety of approaches, highlighting the complexity of the dataset. Ablation studies revealed that the incorporation of multi-modal data and fine-grained annotations improved AQA performance, offering valuable insights for future research endeavors.
In terms of ethical and social implications, though specific negative impacts were not explicitly addressed, the researchers adhered to ethical review guidelines and ensured the paper met the necessary requirements. The study's consent procedures, absence of personally identifiable information, and absence of offensive content were carefully considered, reflecting a responsible approach to data collection and use.

**Additional Feedback:**

Although LucidAction is a large dataset in the AQA field, it is still small compared with other research fields. It is hoped that the author can continue to collect more datasets of different types of sports to enrich LucidAction.

**Documentation:**

Everything is OK.

**Limitations:**

The authors has already addressed the limitations of their work.
Due to the lack of variation in the scenarios in LucidAction, the trained model may be biased, which is not conducive to the fairness of the game.

**Opportunities For Improvement:**

1. Despite the comprehensive multimodal information in the LucidAction dataset, the scale, diversity, or coverage of specific scenarios may still be limited. This means that the generalization ability of models on certain specialized domains or types of actions might be constrained.
2. Although the dataset includes detailed penalty annotations provided by professional gymnasts, this does not guarantee complete coverage of all possible sporting events or skill levels. Consequently, when evaluating action quality in unseen actions or new domains, the performance of the model might deteriorate.
3. While rich in information, the annotation and collection process of the dataset may introduce biases, such as subjectivity from annotators or technical limitations during data acquisition, potentially affecting the accuracy and reliability of the final model.
4. Given that potential negative societal impacts were not discussed in the research, it could imply a deficiency in addressing the ethical and social responsibilities of algorithms.

**Relation To Prior Work:**

yes

**Summary And Contributions:**

This paper introduces a multi-view, multi-level action quality assessment dataset named LucidAction, which aims to address current limitations in AQA research such as reliance on single-modality data, lack of generalizability, and comprehensiveness. The LucidAction dataset is constructed based on the principles of curriculum learning and includes eight different sports, four skill levels, and is applicable to various athletic abilities. It encompasses multi-modal data, including multi-view RGB videos and 2D and 3D pose sequences, ensuring precise capture of complex movements through a high-precision multi-view motion capture system. The data is meticulously annotated by professional gymnasts, including detailed deductions, to establish a solid benchmark annotation. The experiments mentioned in 4.1 and ablation studies mentioned in 4.2 reveal the dataset's complexity and demonstrate the advantages of multi-modal data and fine-grained annotations, providing insights for improving AQA performance.

---

> ### Author Rebuttal · Authors · 2024-08-16
>
> Thank you for providing a comprehensive and thoughtful review of our paper. We are grateful for
> your acknowledgment of the strengths of LucidAction and its contribution to the AQA field.
>
> **Point-by-point response**
>
> >1. The dataset was collected in controlled environments, which may limit its applicability to real-world
> scenarios with varying lighting, backgrounds, and other environmental factors.
>
> The controlled environment was crucial to ensuring a well-organized data collection process, yielding
> high-quality, consistent multi-view data in line with our curriculum concept. While the generalizability
> of models is influenced by the data they are trained on, our method itself is highly adaptable and can
> be applied to various input modalities.
>
> >2. Despite annotations by professional gymnasts, subjective biases during scoring may still exist.
>
> We have taken steps to reduce potential subjectivity by not only involving professional gymnasts,
> referees, and coaches, but also using unified scoring criteria according to the standards, we acknowl-
> edge that some level of subjectivity is inherent in any manual annotation process. To further mitigate
> bias, as noted in L156-L161, we include multiple annotators to repeatedly evaluate each action
> segment.
>
> >3. The dataset currently focuses on gymnastics events, and expanding to other sports would further
> enhance its applicability. Although the dataset includes detailed penalty annotations provided by
> professional gymnasts, this does not guarantee complete coverage of all possible sporting events or
> skill levels. Consequently, when evaluating action quality in unseen actions or new domains, the
> performance of the model might deteriorate.
>
> We recognize the importance of expanding to other sports to enhance its applicability and generalization across a broader range of scenarios. Our method is designed to be broadly applicable, and the
> curriculum-based organization of our data aligns with general principles of sports progression, making
> it extendable to other athletic domains. Expanding LucidAction to include additional sports and
> more curriculum levels is a priority for future work. We believe that ensuring a more comprehensive
> coverage will better support the AQA task in new domains.
>
> >4. Given that potential negative societal impacts were not discussed in the research, it could imply a
> deficiency in addressing the ethical and social responsibilities of algorithms.
>
> Thanks for you suggestions. We will include a discussion covering issues such as potential biases in
> AI-driven evaluations and the importance of fairness and transparency in algorithmic decision-making.
>
> >5. Although LucidAction is a large dataset in the AQA field, it is still small compared with other
> research fields. It is hoped that the author can continue to collect more datasets of different types of
> sports to enrich LucidAction.
>
> We believe LucidAction is a large dataset with richest modality and finest procedure-aware score
> annotation in AQA field. In future work, we are committed to expanding LucidAction by collecting
> more data across a wider variety of sports and scenarios to further enhance the dataset’s utility and
> ensure that it continues to serve as a valuable resource for the AQA community.

---

### Official Review · Reviewer_573E · 2024-07-24
**Multiview, Multimodal Dataset for Action Quality Assessment**

**Rating:** 7
**Confidence:** 5
**Correctness:** Yes.

**Review:**

1. This paper presents a multiview, multimodal dataset for the task of Action Quality Assessment (quantifying how well an action is executed). Dataset covers 8 sports related to gymnastics. Authors capture the dataset using special in-house multicamera setup. Such real multiview dataset would be very valuable to the field AQA.


2. Quite some details are missing and improvements in writing:
- L140-144: Are you collecting 2D or 3D groundtruth on the subset. Please make it clear that you are obtaining 2D groundtruth.
- L140-144: What was the accuracy of the automatic 3D pose estimation pipeline?
- It is unclear how did you collect the action sequences? Did you recruit gymnasts? How did you decide their expertise level? Please provide this info.
- Authors have included a lot of secondary details while not providing the primary info like the above. A good dataset paper should contain all such info.
- L156-161: narrative keeps jumping from one thing to another. Writing needs to be improved.
- L159: Typo: can access to -> can access
- L160-161: How/from where did you obtain perfect action footage? How did you ensure it is perfect?
- Writing can use some work. Writing at quite some places is long-winded. It can be made concise and clear/less confusing.
- In Eq. 1 what is \bigoplus ?
- Table 2: why models perform slightly poorly on LucidAction compared to other datasets? Since the dataset is collected while ensuring high quality, I would have expected models to perform better on Lucidaction dataset than other datasets.
- L241 it is very unlikely that muscle deformations would be captured...any evidence that muscle deformations are being captured?
- L244: please give examples of kinematic features.
- L249-257: limited exploration of curriculum learning but maybe useful.

I feel like strengths of this work outweigh shortcomings. So, my initial rating is accept conditional on that authors do the revisions mentioned above.

**Strengths:**

This paper presents a multiview, multimodal dataset for the task of Action Quality Assessment (quantifying how well an action is executed). Dataset covers 8 sports related to gymnastics. Authors capture the dataset using special in-house multicamera setup. Such real multiview dataset would be very valuable to the field AQA.

**Additional Feedback:**

Please refer to Review section.

**Clarity:**

Writing can be revised and missing info can be added as mentioned in the Review section.

**Documentation:**

Yes.

**Ethics:**

No.

**Limitations:**

Yes, discussed in the paper.

**Opportunities For Improvement:**

Please refer to Review section.

**Relation To Prior Work:**

Overall, okay. In L180: I think credit should be given Jain et al. theirs was the first work to propose reference based scoring model.

Jain, Hiteshi, Gaurav Harit, and Avinash Sharma. "Action quality assessment using siamese network-based deep metric learning." IEEE Transactions on Circuits and Systems for Video Technology 31.6 (2020): 2260-2273.

**Summary And Contributions:**

This paper presents a multiview, multimodal dataset for the task of Action Quality Assessment (quantifying how well an action is executed). Dataset covers 8 sports related to gymnastics. Authors capture the dataset using special in-house multicamera setup. Such real multiview dataset would be very valuable to the field AQA.

---

> ### Author Rebuttal · Authors · 2024-08-16
>
> We sincerely appreciate the reviewer’s thoughtful evaluation of our paper and the invaluable feedback
> provided. It is gratifying to know that the strengths of our work are recognized as outweighing
> its shortcomings. We have carefully considered your comments and made improvements to the
> manuscript. If any of our responses need clarification or if new questions arise, please don’t hesitate
> to reach out. We are committed to addressing any concerns promptly. Thank you again for your
> valuable insights.
>
> **Point-by-point response**
>
> >1. L140-144: Are you collecting 2D or 3D groundtruth on the subset. What was the accuracy of the
> automatic 3D pose estimation pipeline?
>
> Regarding 2D pose estimation, we used a hybrid approach involving both algorithms and human
> review in three stages:
>
> (1) We employed RTMpose [1] pretrained on 7 public datasets including
> MS COCO [2] to estimate 2D poses from single-view videos followed by human quality checks.
> Estimated 2D on some action categories failed human review due to their rare appearance in the
> pretraining datasets;
>
> (2) We manually annotated 2D poses of these rare actions, fine-tuned the
> RTMpose model, and re-estimated the 2D poses, which were then reviewed again;
>
> (3) Any 2D poses
> that still failed the review were manually annotated. This approach balances automated efficiency
> with human validation to ensure accurate 2D pose groundtruth.
>
> For 3D pose estimation, we reconstructed 3D poses using multi-view 2D poses as groundtruth, a
> common method in creating 3D pose datasets [3-6]. Reconstructed 3D pose from multi-view
> 2D are accepted as groundtruth in tasks like human action recognition [7] and motion prediction [8].
> Follow these works, we assess that the accuracy of our 3D poses reconstruction pipeline is sufficient
> for the AQA task. We will add the above information in section 3.1 of the manuscript.
>
> >2. It is unclear how did you collect the action sequences?
>
> We conducted a five-month data capturing during professional gymnastics training courses organized
> according to the standards (L113-114) at a sports university. We will include additional details in
> section 3.2 of the manuscript.
>
> >3. L160-161: How/from where did you obtain perfect action footage? How did you ensure it is
> perfect?
>
> In our work, perfect action refers to penalty-free samples according to the criteria of the standards.
> If no perfect action is captured during regular training sessions, specialized gymnasts will perform
> additional recordings to ensure each action category includes a perfect sample.
>
> >4. L156-161: narrative keeps jumping from one thing to another. L159: Typo.
>
> Thank you for catching this typo. We will correct it in the revised version of the paper and improve
> the writing, ensuring that the discussion follows a logical sequence and is easy to follow.
>
> >5. In Eq. 1 what is \bigoplus?
>
> \bigoplus denotes the operation to fuse the target and exemplar’s representations after the action decoder. In
> experiments we use concatenation following previous work TPT. We will add the clear explanation of this symbol in
> the revised version.
>
> >6. Table 2: why models perform slightly poorly on LucidAction compared to other datasets?
>
> We believe the slightly lower performance on LucidAction stem from its increased complexity.
> Unlike previous datasets like MTL-AQA and FineDiving, whose data source is competition footage,
> LucidAction was collected intensively based on curriculum, resulting in many similar action samples
> with closely related scores. This demands finer-grained distinctions between actions, making the AQA
> task more challenging and slightly lowering performance metrics. Additionally, absolute performance
> variations across sports are common, as seen in the performance of previous work TPT on AQA-7,
> where ρ reached 0.8969 on diving subset and only 0.6965 on skiing.
>
> >7. L241 any evidence that muscle deformations are being captured?
>
> Although muscle deformations is observed in some frames from RGB videos in LucidAction, we
> agree that capturing the deformations can not be ensured with the current setup in all data samples.
> Our intention was to convey that pose data compared to video may lose critical appearance-related
> information relevant to the AQA task. We will revise this statement to reflect the respective advantages
> of pose and video modalities more accurately.
>
> >8. L244: please give examples of kinematic features.
>
> We apologize for the imprecise phrasing. We meant kinematic properties such as keypoint movement
> velocity, displacement distance, angles, etc.
>
> >9. In L180: I think credit should be given Jain et al. theirs was the first work to propose reference based scoring model.
>
> Thank you for this correction. We will acknowledge Jain et al.’s contribution to the development
> of reference-based scoring models in the revised manuscript.
>
> **Reference**
>
> [1] Jiang, et al. Rtmpose: Real-time multi-person pose estimation based on mmpose, arXiv 2023.
>
> [2] Lin, et al. Microsoft coco: Common objects in context. ECCV 2014.
>
> [3] Shahroudy, et al. Ntu rgb+d: A large scale dataset for 3d human activity analysis. CVPR 2016.
>
> [4] Liu, et al. Ntu rgb+d 120: A large-scale benchmark for 3d human activity understanding. TPAMI 2020.
>
> [5] Cai, et al. HuMMan: Multi-modal 4d human dataset for versatile sensing and
> modeling. ECCV 2022.
>
> [6] Dong, et al. Fast and robust multi-person 3d pose estimation and tracking from multiple views. TPAMI 2021.
>
> [7] Wang, et al 3mformer: Multi-order multi-mode transformer for skeletal action
> recognition.CVPR 2023.
>
> [8] Xu, et al. Auxiliary tasks benefit 3d skeleton-based human motion prediction. ICCV 2023.

---

> > ### Comment · Reviewer_573E · 2024-08-21
> >
> > Thanks to the authors for answering my questions/concerns. Overall, I think the dataset is good and a valuable contribution and can also help enhance new SOTA paradigms like Neurosymbolic AQA approaches. Therefore, I am voting to accept this paper.

---

### Official Review · Reviewer_oQZi · 2024-07-26
**LucidAction: A Hierarchical and Multi-model Dataset for Comprehensive Action Quality Assessment**

**Rating:** 6
**Confidence:** 5
**Correctness:** Yes
**Clarity:** Yes

**Review:**

**Quality:**
- The quality of this research is commendable, particularly in the construction of the LucidAction dataset.
- Meticulous annotation and inputs from professional gymnasts ensure high-quality data.
- Integration of multi-modal data (RGB videos, 2D/3D pose sequences) enhances dataset utility and reliability.

**Clarity:**
- The presentation is clear and well-structured.
- Hierarchical organization of LucidAction across sports and skill levels is effectively communicated.
- Experimental methodologies and evaluation metrics are clearly defined, ensuring transparency and reproducibility.

**Originality:**
- LucidAction is original in its approach to AQA, leveraging curriculum learning and multi-angle data capture.
- Surpasses existing datasets with its diversity and sophistication in data collection.

**Significance:**
- Significant impact on sports analytics and computer vision.
- Supports a wide range of athletic skills, fostering advancements in AI-driven performance evaluation.
- Open-sourcing promotes collaboration and accelerates research progress.

### Pros

- Innovative use of curriculum learning for dataset construction.
- Inclusion of multi-modal data enriches dataset quality.
- Detailed annotation ensures benchmark reliability.
- Clear presentation and transparent methodologies.
- Potential to advance AQA capabilities across sports disciplines.

### Cons

- Dependency on high-quality motion capture systems.
- Limited accessibility in resource-constrained environments.
- Initial dataset scope could be expanded for broader applicability.
- Missing references: Anti-UAV: A large multi-modal benchmark for UAV tracking, Understanding humans in crowded scenes: Deep nested adversarial learning and a new benchmark for multi-human parsing, Anti-uav410: A thermal infrared benchmark and customized scheme for tracking drones in the wild

**Strengths:**

- Innovative use of curriculum learning for dataset construction.
- Inclusion of multi-modal data enriches dataset quality.
- Detailed annotation ensures benchmark reliability.
- Clear presentation and transparent methodologies.
- Potential to advance AQA capabilities across sports disciplines.

**Additional Feedback:**

N/A

**Documentation:**

N/A

**Limitations:**

See Review

**Opportunities For Improvement:**

See Review

**Relation To Prior Work:**

Yes

**Summary And Contributions:**

The research addresses challenges in Action Quality Assessment (AQA) due to limited, single-mode datasets from isolated competitions, which constrain the generalizability of AQA models. To overcome these issues, the authors introduce LucidAction, a novel multi-view dataset structured on curriculum learning principles. LucidAction includes a hierarchical structure covering eight sports events with four curriculum levels, facilitating progressive skill acquisition across diverse athletic abilities. It incorporates multi-modal data such as RGB video and 2D/3D pose sequences, captured with a precise Motion Capture (MoCap) system. The dataset is meticulously annotated with penalties from professional gymnasts, establishing robust ground truth annotations. Experimental evaluations using various regression baselines reveal the dataset's complexities, while ablation studies highlight the benefits of multi-modal data and detailed annotations for enhancing AQA performance. The authors plan to openly release the dataset and code to foster advancements in AI applications in sports.

-	The paper introduce LucidAction, the first AQA dataset structured according to the principles of curriculum learning
-	Through experimentation, the paper investigate the effectiveness of multi-model inputs and fine-grained hierarchical annotations in enhancing AQA performance, thereby offering insights into methodological advancements for the field.

---

> ### Author Rebuttal · Authors · 2024-08-16
>
> We sincerely appreciate your insightful review and the recognition of the strengths in our work. Your detailed feedback on the quality and significance of the LucidAction dataset is invaluable.
>
> **Point-by-point response**
>
> >1. Dependency on high-quality motion capture systems. Limited accessibility in resource-constrained environments.
>
> Our approach leverages sychronized multi-view videos, which may require specialized hardware. However, our work is the first to provide extra track on data modality for AQA task. It provides new solution and bring a significant performance improvement to AQA task when taking 3D poses and RGB videos as dual-stream input (see Table 3(b)). Importantly, our dataset and method are still compatible to use single-view RGB or 2D pose data as input when multi-view settings are not available, offering a trade-off between hardware investment and performance. This flexibility allows for wider applicability in resource-constrained environments.
>
> >2. Initial dataset scope could be expanded for broader applicability.
>
> We appreciate your suggestion to expand the scope of the dataset. The construction pipeline of LucidAction could generalize to other activities, and the curriculum-based organization of data from simple to complex aligns with general principles of sports. While the current version of LucidAction covers eight events, we recognize the potential for including additional sports and activities in future work.
>
> >3. Missing references: Anti-UAV: A large multi-modal benchmark for UAV tracking, Understanding humans in crowded scenes: Deep nested adversarial learning and a new benchmark for multi-human parsing, Anti-uav410: A thermal infrared benchmark and customized scheme for tracking drones in the wild
>
> Thank you for pointing out the missing references. We will carefully review these works and include relevant citations in our paper.

---

### Official Review · Reviewer_XN6L · 2024-07-26
**Really useful benchmark for AQA and related fields**

**Rating:** 8
**Confidence:** 4

**Review:**

pros
The quality of the manuscript and significance of the work are high.
Sports actions are challenging for pretrained models due to their extreme nature. This dataset and the insights presented in the paper will be significant for the field of sports action recognition and recostruction.

cons
I think the authors could improve the flow and language of parts of the text for even better readability and understanding

**Strengths:**

The number and expertise of annotators (coaches and athletes) is remarkable
The number of views, mocap system, number of sports is remarkable

**Additional Feedback:**

NA

**Clarity:**

The paper is well written, alas a lot of adjectives that complicate the text without adding to the context.

**Correctness:**

The claims are correct.

One question I have is regarding the claim that 'The input data is mono-modal, consisting of video captured by a single moving camera.'
I believe this is often not the case, especially for high-profile events like the Olympics, where multiple cameras capture different angles.

**Documentation:**

Yes.

**Ethics:**

The authors mention that they received consent from the athletes and have anonymized their faces.

**Limitations:**

The authors have reported limitations.

**Opportunities For Improvement:**

- The abstract (and parts of the text) sound too GPT-written, maybe try to remove some of the dramatic adjectives
- Does the Bengio quote belong in the middle of the regular text?
- I think the authors mean multi-modal and not multi-model

**Relation To Prior Work:**

Yes.

**Summary And Contributions:**

The authors collected and annotated an extensive multiodal dataset of 8 sports, 8 camera views, and MoCap data.

The authors also conducted experiments on the impact on performance of different contrastive regression workflows, multi-modal inputs, and the introduced here hierarchical annotations on their dataset and related datasets.

---

> ### Author Rebuttal · Authors · 2024-08-16
>
> Thank you for the constructive review of our paper. We are grateful to hear that you found our work significant for the field of sports action recognition and reconstruction. We will implement the suggested revisions to improve the clarity and readability of our paper.
>
> **Point-by-point response**
>
> >1. The authors could improve the flow and language of parts of the text for even better readability and understanding.
>
> Thank you for the advise, we will carefully revise the abstract and other relevant sections to reduce the use of adjectives and ensure that the language is more direct and focused on conveying the core contributions of the paper.
>
> >2. Does the Bengio quote belong in the middle of the regular text?
>
> The quote was intended to provide context for the discussion, but we understand that its placement may be disruptive and will reconsider to put it into regular text.
>
> >3. I think the authors mean multi-modal and not multi-model.
>
> Thank you for pointing this typo out. We meant multi-modal and will correct this in the manuscript.
>
> >4. One question I have is regarding the claim that 'The input data is mono-modal, consisting of video captured by a single moving camera.' I believe this is often not the case, especially for high-profile events like the Olympics, where multiple cameras capture different angles.
>
> We apologize for the confusion. While multi-camera setups are common in high-profile events, previous AQA datasets have relied on single-view footage [1,2,3] or combined replays from different angles without temporal synchronization [4], due to the constraints of publicly available competition footage. As a result, it is unable to build temporally aligned multi-view inputs like LucidAction provides in previous datasets.
>
> **References**
>
> [1] Parmar, et al. "Action Quality Assessment Across Multiple Actions." WACV 2019.
>
> [2] Parmar, et al. "What and How Well You Performed? A Multitask Learning Approach to Action Quality Assessment." CVPR 2019
>
> [3] Xu, et al. "FineDiving: A Fine-grained Dataset for Procedure-aware Action Quality Assessment." CVPR 2022.
>
> [4] Liu, et al. "A Figure Skating Jumping Dataset for Replay-Guided Action Quality Assessment." MM 2023.

---

### Author Rebuttal · Authors · 2024-08-16

**Main Response**

Dear reviewers, thank you for your thoughtful feedback and suggestions for improvement. We are
glad to see the overall positive reception of our work. We have provided individual responses to each
reviewer. Below is a summary of the revisions made in response to your feedback.

Reviewer XN6L noted that multi-view setups are common in high-profile events like the Olympics.
While this is true, previous AQA datasets have typically relied on single-view footage or lacked
temporal alignment across multiple cameras due to the limitations of publicly available competition
footage. LucidAction addresses this gap by providing synchronized multi-view data, offering a more
comprehensive resource for AQA tasks.

Reviewer oQZi expressed concerns about the accessibility of our multi-view setup in resource-
constrained environments. While the multi-view approach does require additional hardware, it offers
significant benefits when such setups are feasible, as demonstrated by our experimental results.
Importantly, our dataset and methods are also compatible with single-view RGB or 2D pose data,
allowing for flexibility and a trade-off between hardware investment and performance.

Reviewer 573E asked how we collect the action sequences and on which subset do we collect
groundtruth 2D pose/3D pose. We conducted a five-month data capture during professional gymnastics training courses, organized by curriculum levels. We employed an algorithm-human hybrid
pipeline to ensure the accuracy of 2D and 3D ground truth data, and we will add this information to
the manuscript. We also addressed Reviewer FFMn’s concerns about potential biases in annotations
by involving multiple professional gymnasts and using unified scoring criteria.

Reviewer 573E inquired about the slightly lower model performance on LucidAction compared
to previous datasets. We believe the slightly lower performance on LucidAction stem from its
increased complexity. Unlike previous datasets like MTL-AQA and FineDiving, whose data source
is competition footage, LucidAction was collected intensively based on curriculum, resulting in
many similar action samples with closely related scores. This demands finer-grained distinctions
between actions, making the AQA task more challenging and slightly lowering performance metrics.
Additionally, absolute performance variations across sports are common as seen in the performance
of previous work TPT on different sport subset of AQA-7.

Follow the suggestions from Reviewer XN6L and 573E, we will revise our manuscript to improve
the clarity and readability, ensuring that the language is more direct and focused on conveying the
core contributions. Specific issues, such as typos will be corrected.

Finally, as suggested by Reviewer oQZi and FFMn, we recognized the importance of expanding
LucidAction to include a broader range of sports and activities, and this is a priority for our future
work.
We hope these changes address your concerns, and we appreciate your valuable feedback.

---

### Decision · Program_Chairs · 2024-09-26

**Decision:**

Accept (Poster)

**Comment:**

This paper presents LucidAction, which is a systematically collected multi-view action quality assessment (AQA) dataset structured on curriculum learning principles.  It has a three-tier hierarchical structure, including eight sports events with four curriculum levels. The dataset has multi-modal data, such as multi-view RGB video, 2D and 3D pose sequences, using high-precision multi-view Motion Capture (MoCap) system.  The dataset is meticulously annotated with penalties from professional gymnasts. Evaluations are provided using various regression baselines. The authors have a plan to openly release the dataset and code.

All of the four reviewers are positive towards the acceptance of this paper. Rebuttal addresses some of reviewers' concerns. AC agrees with reviewers to accept this paper.